# An Increased Risk of Stunting among Newborns in Poorer Rural Settings: A Cross-Sectional Pilot Study among Pregnant Women at Selected Sites in Rural Cambodia

**DOI:** 10.3390/ijerph16214170

**Published:** 2019-10-29

**Authors:** Margit Steinholt, Sam Ol Ha, Chandy Houy, Jon Øyvind Odland, Maria Lisa Odland

**Affiliations:** 1Helgelandssykehuset, 8801 Sandnessjoen, Norland, Norway; 2Department of Public Health and Nursing, Norwegian University of Science and Technology, 7491 Trondheim, Norway; jon.o.odland@ntnu.no (J.Ø.O.); m.l.odland@bham.ac.uk (M.L.O.); 3Trauma Care Foundation, Battambang, Cambodia; olliekhm@gmail.com (S.O.H.); chandyhouy@gmail.com (C.H.); 4Institute of Applied Health Research, University of Birmingham, Birmingham B15 2TT, UK

**Keywords:** persistent toxic substances, stunting, low income settings, Cambodia

## Abstract

We conducted an observational study of 194 pregnant women from two different study sites in rural Cambodia. Socioeconomic and anthropometric data was obtained from the women and their newborns. In addition, we collected blood and urine samples from the women for further analyses in planned papers. There were significant differences between the two study groups for clinical outcomes. The mothers from the poorer area were shorter and weighed less at the time of inclusion. Their babies had significantly smaller head circumferences and a lower ponderal index. Conclusion: There are significant anthropometric differences between women and their newborns from two separate study sites in Cambodia. Possible associations between stunting and exposure to Persistent Toxic Substances (PTS) as organochlorines and toxic trace elements will be investigated in future studies.

## 1. Introduction

Cambodia is a small country in South East Asia with a population of about 16 million people. It is ranked as a low-income country. Cambodia’s Gross National Product (GNP) per capita is 1615 USD and one third of the population is defined as extremely poor; living on less than 1.25 USD per day [1].

In the last few decades, Cambodians have, in addition to natural disasters like flooding and draught, experienced numerous man inflicted catastrophes with unstable and insufficient food supply. During the Vietnam war (1965–1973), the US military dropped 2,756,941 tons of bombs over the country’s rice fields [2]. The Khmer Rouge regime ruled the country from 1975 to 1979, and even though the rice harvests were abundant during these years, the crops were mainly exported, and an unknown number of people died of starvation and illnesses. A devastating civil war raged for almost 20 years following the Vietnamese invasion in 1979, forcing thousands of people into internal displacement. Landmines and unexploded ordnances (UXOs), remnants from more than 60 years of war and conflicts, still threaten the rural population and make cultivation and farming a perilous activity. Poisonous leakage enters groundwater and soil and contributes to invisible health hazards. In recent years Cambodian authorities have allowed foreign investors to dump toxic substances in rural areas in return for investments in infrastructure like roads and hydroelectric powerplants. The expanding garment industry along the waterways of Phnom Penh has few restrictions concerning waste regulations, and Cambodia is a welcoming tax haven for international companies.

Annually, an estimated 9 million people in the world die prematurely due to pollution of air, soil, and water. The numbers are three times higher than fatalities from AIDS, malaria, and tuberculosis combined, and the vast majority (92%) of the premature deaths occur in low-middle income countries. Despite overwhelming evidence that unhealthy living and working conditions are closely related to adverse health outcomes, pollution mitigation and control have not been considered as public health issues. The Lancet commission on pollution and health suggests that these policies must change [3].

There is, however, increasing global awareness about the negative health impacts of pollutants, including persistent toxic substances (PTS). Organochlorines (OCs) and certain toxic trace metals as arsenic, cadmium, cobalt, lead, and mercury are under particular scrutiny [4,5]. Since 1991, the Arctic Monitoring and Assessment Programme (AMAP) has conducted numerous studies in different parts of the world investigating PTS and their impact on pregnancy and early childhood. In 2015 the AMAP Human Health Assessment Report presented substantial evidence for adverse neurobehavioral effects, immunologic effects, reproductive effects, cardiovascular effects, endocrine effects, and carcinogenic effects on human health [6].

The aim of this study is to determine the pregnancy outcome, socioeconomic background, and anthropometric data possibly associated with the level of some selected chemical and toxic hazards among pregnant women and their offspring in two rural communities in Cambodia. The study sites were selected because despite their relative proximity, the two populations have different ways of living and also different food consumption patterns. This paper will assess the socioeconomic and anthropometric data for the study groups.

## 2. Materials and Methods

### 2.1. Study Design

Our study is an observational study compatible with previous studies done in the context of AMAP [6,7]. The study is part of the global assessment conducted by researchers affiliated with AMAP and researchers from the eight Arctic countries; Canada, Denmark/Greenland, Finland, Iceland, Norway, Russia, Sweden, and the United States of America. This paper describes the framework for the study; including the two study areas, the anthropometric and socio-economic findings from the recruited women, data on the newborns and some potential sources of exposure.

### 2.2. Study Setting

Two rural areas in the North-western part of Cambodia were selected for our study, as shown in Figure 1 [8].

Area 1, Chroy Sdao commune, is located approximately 30 km north of the provincial center, the city of Battambang, and has a population of 21,000 people. Area 1 is mainly an agricultural community. During the past 20 years, rice farming has become multi-seasonal, meaning farmers plant and reap their crops at least twice a year. There is extensive use of fertilizers, herbicides, pesticides, and insecticides. All villages are accessible both in the dry and in the rainy season. The infrastructure is adequate with roads, shops, schools, and primary health care in close proximity for most inhabitants. Two health centers provide basic antenatal care and delivery services for uncomplicated births.

Area 2 is located in the Eak Phnom district. The district is also called “the floating villages” since the majority of the population, 18,000 people, spend most of their lives in boats and rafts on the waterways between the cities of Battambang and Siem Reap. More than 90% of the population make their livelihood from fishing. The way of living is semi nomadic; the families follow the fish during the different seasons; simply bringing their homes and children with them. The landscape changes profoundly in the rainy season when flooding of the Mekong river reverses the flow in the Tonle Sap river. This yearly phenomenon increases the surface of the Tonle Sap lake from 2600 km^2^ to 16,000 km^2^. The lake and its tributaries add 10–15 meters to the water levels, and fisheries are at their peak. An estimated 300,000 tons of fish are caught every year, providing an important source of proteins for the populations as well as much needed income [9]. The two health centers in Eak Phnom district are only accessible by boat, and when the lake expands during the rainy season, the travel distances also increase. Both health centers provide basic antenatal care and delivery service for uncomplicated births. There are few schools, and grocery shopping can only be done by traveling some distance by boat.

### 2.3. Inclusion

The study recruited 194 women and their babies; 120 women from the Chroy Sdao commune, area 1, and 74 from the Eak Phnom district, area 2. The inclusion period for the women lasted from October 2015 through May 2016. It was challenging to recruit, especially from area 2 (the floating villages), due to the seasonal shifts. In the dry season with low water levels, it is difficult to access the villages. In the rainy season the families have gone fishing, and it may take days to track them down. Informed written consent was given by all the participants, including thumb prints by those who are illiterate. All women were in the last trimester of their pregnancy, the majority in gestational week 30–32.

There is missing data on 45 deliveries; eight of these are women are from area 2, the floating villages. The remaining 37 are women from the inland areas who, according to their own or the local midwives’ information, left their locations just prior to birth to go to Thailand for employment. We have data for 149 neonates; 83 from area 1 and 66 from area 2.

### 2.4. Data Collection and Analyses

The women were examined and interviewed by the presiding midwife and local researchers. The questionnaire was similar to those used in comparable AMAP-studies, however, it was translated into Khmer and adapted to and with additional questions adjusted to the Cambodian context. The gathered information included name, age, height, and weight at the time of inclusion, reproductive history, and socio-economic background. The questionnaire also documented life-style, environmental history, and consumption frequencies of the most commonly eaten food. Appendix A has further information about the questionnaire.

At the time of birth, a standard form was completed by the medical personnel in the maternity units. The data included mode of delivery, gender of the baby, APGAR score, weight, length, and head circumference. Any complications for the mother or malformations and stillbirths were also noted.

### 2.5. Sampling and Analyses

Maternal blood and urine samples were obtained at the time of registration. We followed the same procedures as those used in similar studies conducted in the AMAP context [6].

All biological material and written information were stored at TCF-Cambodia’s office in Battambang. The blood samples were transported from Battambang to the Pasteur Institute, Nha Trang in October 2016 for further analysis.

### 2.6. Ponderal Index

The most common method to estimate fat levels in individuals is the use of Body Mass Index (BMI); BMI = weight (kg)/height (m)^2^. BMI is a useful tool in adults, however, for children it has not proven to be as accurate due to the different fat distribution in children during various stages of growth. We have used the ponderal index (PI) to estimate the nutritional status for the newborns in our study; PI = Weight (Kg)/ Height (m)^3^. The ponderal index provides valid results even for very short and very tall individuals, and hence gives more reliable information about growth retardation in babies than the body mass index (BMI) [10].

Intrauterine growth restriction can occur at any stage in a pregnancy and will consequently give different outcomes for the new-born; the baby may experience symmetrical growth or asymmetrical growth. A longstanding growth restriction, especially if inflicted early in fetal life, will most likely lead to symmetrical reduction in both weight, length, and head circumference. The ponderal index will appear to be normal despite the baby being small for its gestational age (SGA). In contrast, more acute growth restriction happening late in the pregnancy will result in a normal head circumference, possibly some length impairment, but mostly a reduction in weight. Thus, the SGA is asymmetrical, and the ponderal index will also be low [11,12].

### 2.7. Statistical Analyses.

Statistical analyses were processed with SAS JMP^®^ (SAS Institute, Kerry, NC, USA). Continuously and symmetrically distributed variables are expressed as mean values with 95% confidence intervals (95% CIs) compared with the students t-test.

Categorical variables are presented in contingency tables with 95% confidence intervals (CI) for a two-tailed comparison. Confidence interval analysis were used to compare proportions and means, and differences were considered significant if 95% CI for the difference did not include zero or the 95% CIs of the two groups were not overlapping. *p*-values <0.5 were considered.

### 2.8. Ethical Approval

The forms and written files of demographical and laboratory data were stored in a locked room at the Trauma Care Foundation head office in Battambang, Cambodia. Access to non-anonymous data was restricted to members of the research team. The data was stored and processed according to approved National Ethics Committee for Health Research of the Ministry of Health, Cambodia (ref. 0365 N.E.C.H.R., 29/12/2014 and 114 N.E.C.H.R; 28/03/2016). The study was approved by the Regional Ethics Committee North, reference 2105/2486.

## 3. Results

### 3.1. Sociodemographic Characteristics

The sociodemographic, anthropometric, and medical information about the women and the neonates can be found in Table 1 and Table 2.

#### 3.1.1. Women

There was no difference in mean parity or age (mean 26.6 SD 6.2) between the two study areas. Eighty-two women were expecting their first baby while 93 had one to three deliveries prior to this pregnancy. Nineteen women had given birth to four or more children, and 14 of the multiparas lived in the floating area.

The women from area 2 are significantly shorter (152.6 cm versus 156.9 cm) than the women from area 1. They also weighed less (56.4 kg versus 60.2 kg) at the time of inclusion, as can be seen in Table 1. 

As seen in Table 1, the mean BMI for the mothers is the same in both groups. Table 2 shows that the reported gestational duration of the pregnancy is 10 days shorter (38.4 weeks versus 40.0 weeks) in area 2 compared to area 1, however, no premature (<37 gestational week) births are recorded.

We have data from 149 deliveries, and the findings are presented in Table 2. The majority of the women gave birth with a midwife at the local health centers. All operative deliveries (vacuum and caesarean sections) were performed on women from area 1. No complications were reported from the 66 deliveries in area 2.

There are important differences in socio-economic variables between the two study areas. The women from the inland areas have more formal education, as the vast majority (90%) had completed at least grade 9 or more. Ten had a high school diploma. In contrast, among the women from the floating villages only 48 out of 74 (65%) had attended primary school (up to grade 9), and none had any formal education beyond that.

#### 3.1.2. Neonates

We have information about gender for 172 babies, 81 boys (47 %) and 91 girls (53%), in Table 2. This number is higher than the recorded number of births (149); the reason being that some babies were born at home, and the midwife then recorded the gender at the time of the first vaccination. For 146 neonates, 64 from the river and 82 from the inland population, we have data about weight, length, and head circumference at the time of birth. All babies were born alive, and 130 neonates had also been assessed according to the Apgar score. The average was score 9 five minutes after birth. There was no difference in Apgar score between the two groups. The babies were born around the expected term, however, the women from the river area had significantly shorter pregnancies than inland; 38.4 weeks compared to 40.0 weeks, Table 2.

The babies from area 2 weighed less (3000 g vs. 3200 g) and had significantly lower ponderal index than the newborns from area 1; ponderal index (kg/ m3) being 24.7 versus 28. 6, Table 2.

The head circumference was also significantly smaller in the babies from area 2 (30.2 cm vs. 32.6 cm).

### 3.2. Possible Sources of Exposure to Persistent Toxic Substances:

Persistent toxic substances can be found almost everywhere in the surroundings of human activities. In Table 3, we present some potential sources of exposure to organochlorines and toxic trace metals to our study population.

#### 3.2.1. Drinking Water

Most of the women, (93%), in the floating areas reported that the river was the family’s main source of drinking water. Five bought bottled water for drinking and cooking. In comparison only seven (5%) of the 119 women from the inland used rivers or small man-made ponds for drinking water. The majority (90%) collected rainwater (64/119) or had access to wells (44/119), Table 3.

#### 3.2.2. Insecticides

Almost 80% of the households from the river villages used insecticides, and the majority only sprayed their homes. In comparison 40% of the families in the inland areas used insecticides with only 12% spraying their homes.

#### 3.2.3. Fishing Locations

The majority of families (96%) from the floating areas reported that they fish in the big river (Tonle Sap). Less than half (49%) of the women from the inland catch their fish in smaller rivers and/or inland lakes or manmade ponds (not the Tonle Sap).

#### 3.2.4. Additional Food Sources

All women from the inland have access to rice, meat, fruit, and vegetables from their own village. They either grow food themselves or buy from each other or the local market. In contrast, most of the women from the floating villages (93%) need a boat to go a market or have to wait for a “trading boat” to pass by in order to find the same items. The cost for groceries is higher when bought on the river.

## 4. Discussion

We found distinct anthropometrical and socio-economic differences between two groups of pregnant women and their neonates residing less than 100 km apart. While there was no difference in age or parity, the women from the floating villages, area 2, were significantly shorter (4.3 cm) and weighed less (3.8 kg) at the time of inclusion compared to the women from the inland, area 1. Their babies were also smaller with a significantly lower ponderal index and smaller head circumference. These discrepancies were unexpected and have important clinical impacts. The findings must be explored further.

As far as we know, this is the first research of its kind with material from Cambodia. One major strength of our study is the close relationship the Cambodian researchers and fieldworkers have with the local health centers and their staff. This ensured that the women who were included represented a cross section of the pregnant population in the villages at this specific time. Mutual trust also made the answers and information given by the women reliable.

The two study districts are geographically quite close to each other, and all of the women had lived in their present area for most of their lives. We are not aware of any ethnic differences that could explain the discrepancies found in anthropometric measurements between the two groups of women and newborns.

An important limitation is that women who do not attend antenatal care were not included.in the study. Women in Cambodia who never seek antenatal check-ups (ANC) also tend to deliver their child at home. The poorer the woman, the less likely she is to come to the health center for ANC or birth [13]. Especially in the floating areas, we may have missed some potential participants due to poverty or the fact that the families are constantly on the move for fishing purposes.

Another concern is gestational length. None of the due dates were determined or confirmed by trained ultrasound health personnel. In the study we had to rely on the woman’s own estimation of when she expected to deliver. Any random error concerning the term date should be evenly distributed among the informants, independently of location. The difference found in gestational length is most likely real.

The sample size in our study can be argued to be small. However; despite the limited number of participating women and babies, we did find interesting and quite substantial differences in anthropometrics and socioeconomic status between the two populations. This calls for further investigation.

In the Newborn Cross-Sectional Study of the INTERGROWTH-21st Project, Villar et al. have calculated international standards for newborn weight, length, and head circumference by gestational age and sex. The standards are based on data from over 20,000 low risk pregnancies from eight different geographical locations in the world; Brazil, China, India, Italy, Kenya, Oman, the UK, and the US. The study finds few anthropometric differences related to ethnicity in healthy neonates born by healthy mothers living in favorable conditions. According to the study an average term newborn weighs 3300 g (+/−SD 500 g), measures 49,3 cm (+/−SD 1,8 cm) and has a head circumference of 33,9 cm (+/−SD 1,3) [14].

The babies from our study weigh less and have a smaller head circumference than the international standards estimated by the INTERGROWTH-21st Project. The babies are also shorter. The largest discrepancy is found in babies from the river areas (area 2).

Stunting is one of the great challenges in South East Asia as in other low resource settings in the world. Stunting, meaning a reduced height for age, is an indicator of chronic malnutrition, and is associated with impaired cognitive and motor development [15,16]. Impaired growth in the uterus increases the risk for non-communicable diseases later in life [17]. Suboptimal nutrition may lead to irreversible changes in the developing brain and also epigenetic alterations in physical growth. The reduced ability for children to reach their full mental and physical potential is a tragedy for the individual, but also a loss for the communities and countries most affected.

Globally 22% of the children under 5 years old are stunted [18]. Several studies have showed a correlation between a short maternal stature and mortality and growth restrictions in children. Maternal height can therefore be used as a marker to explain intergenerational linkages in health. The height of an adult woman reflects the health stock accumulated in her life time, including social and environmental exposures in her own early childhood [19,20]. Özaltin et al. show in their study from 54 low-and-middle-income countries that a 1 cm increase in maternal height is associated with reduced mortality, stunting, and underweight in the offspring. They further found that maternal stature was a more important determinant in childhood stunting and underweight than both low education and poverty; thus, confirming the intergenerational link [19].

According to the Nutrition Landscape information System, Cambodia has a long history of female malnutrition. From 1980 to 2014 the prevalence of thinness in women in reproductive age dropped from 26% to 14%, however, the prevalence is higher (28%) among adolescent girls. In 2014 the malnutrition for children under 5 years of age showed that 23.9% of the children were underweight while 32.4% were stunted [21].

The most plausible explanation for the significant differences found between the two groups of women and their offspring in our study, especially expressed in maternal height and the size of the neonates, must be that living conditions on the river are harsher and less favorable than the inland. The families are poorer, and the women are less self-sustained financially. The consequences are stunting for both mothers and babies.

The women from the inland are better educated, more often hold formal jobs, and have easier access to food items like rice, fruits, vegetables, and meat. The study population from the inland area also has better access to clean water for drinking and other domestic chores. The majority of the households on the river sprayed insecticides in their homes. Less than half of the families in the inland area used insecticides, and those who did, sprayed the crops and not in the house. Thus, families on the river probably have more ambient chemicals in their immediate surroundings than the inland study population.

Our hypothesis is that the stunting is the result of generations of girls growing up in surroundings unable to meet their needs for sufficient and nutritious food. When these girls themselves become mothers, they pass on the growth restriction to their children.

In the follow-up studies we will look into possible correlations between the anthropometric findings and the levels of persistent toxic substances in the blood samples from the women while pregnant. We already know that disadvantaged populations are more highly exposed to pollutants [3]. That also correlates with our study. Table 3 shows that the poorest women, the informants from the river, had less access to clean drinking water, and also used more chemicals in their homes.

## 5. Conclusions

In a study of 194 women from two low-income settings in rural Cambodia, we found that the poorer women were significantly shorter and gave birth to babies with a significantly smaller head circumference and a lower ponderal index than women from the more advantaged villages. Our findings correlate with other studies showing that stunting is a result of intergenerational influence on maternal stature due to lack of sufficient and nutritious food. As part of our further investigation, we will look into possible associations between stunting and exposure to persistent toxic substances.

## Figures and Tables

**Figure 1 ijerph-16-04170-f001:**
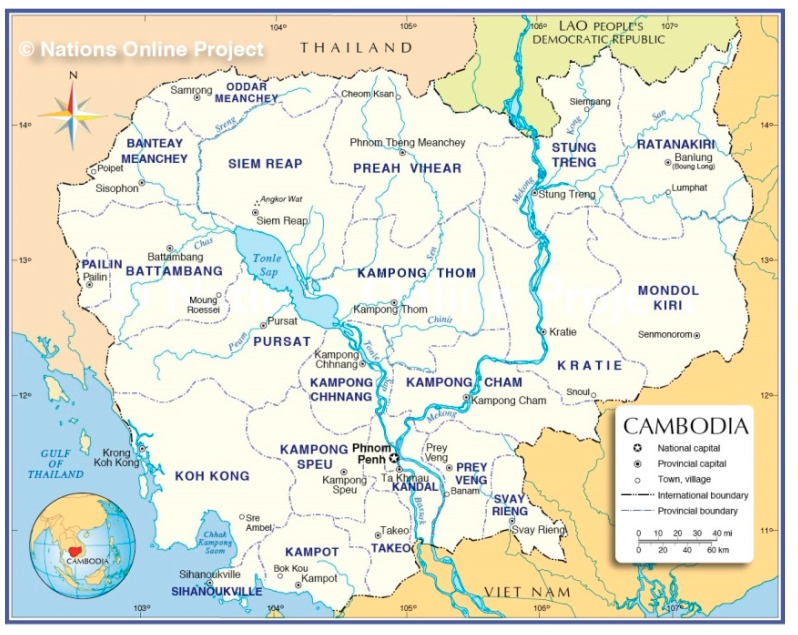
Map of Cambodia

**Table 1 ijerph-16-04170-t001:** Demographic information of the women included in the study.

	Main Variables	Area																		*p*-Value
		All locations						Inland Area (1)						Floating Area (2)						
		Count	Percent	Mean	SD	95% CI		Count	Percent	Mean	SD	95% CI		Count	Percent	Mean	SD	95% CI		
						Lower	Upper					Lower	Upper					Lower	Upper	
	Age	194		26.6	6.2	25.8	27.5	120		26.7	6.5	25.5	27.9	74		26.6	5.7	25.3	27.9	0.94
	Body Weight (kg)	194		58.8	8.1	57.6	59.9	120		60.3	7.9	58.8	61.7	74		56.4	8	54.5	58.2	0.001
	Height (cm)	194		155.3	5.2	156	154.5	120		156.9	4.8	156.1	157.8	74		152.6	4.7	151.5	153.7	<0.0001
	BMI	194		24.4	3	23.9	24.8	120		24.5	2.9	23.9	25	74		24.2	3.2	23.5	24.9	0.55
	Live in Area	194		21	9.9	22.4	19.6	120		21.6	9.2	19.9	23.3	74		20.1	11	17.6	22.7	0.31
Parity																				
	Para 0	82	42.3			35.5	49.3	59	49.2			40.4	58.0	23	31.1		21.7	42.3		
	Para (1–3)	93	47.9			41	54.9	56	46.7			38.0	55.6	37	50.0		38.9	61.1		
	Para (≥4)	19	9.8			6.4	14.8	5	4.2			1.8	9.4	14	18.9		11.6	29.3		
Education																				
	Primary	83	42.8			36	49.8	44	36.7			28.5	45.5	39	52.7		41.5	63.6		
	Secondary	72	37.1			30.6	44.1	63	52.5			43.6	61.2	9	12.2		6.5	21.5		
	No Education	39	20.1			15.1	26.3	13	10.8			6.5	17.6	26	35.1		25.2	46.5		
Occupation																				
	House wife	67	34.5			28.2	41.5	34	28.3			21.0	37.0	33	44.6		33.8	55.9		
	Workers	29	14.9			10.6	20.6	28	23.3			16.7	31.7	1	1.4		0.2	7.3		
	Farmers	55	28.3			22.5	35.1	54	45.0			36.4	53.9	1	1.4		0.2	7.3		
	Teacher	4	2.1			0.8	5.2	4	3.3			1.3	8.3	0	52.7		41.5	63.7		
	Fishing	39	20.1			15.1	26.3	0						39						
Previous child died <5 yrs																				
	No	179	92.3			87.6	95.3	112	93.3			87.4	96.6	67	90.5		81.7	95.3		
	Yes	15	7.7			4.7	12.4	8	6.7			3.4	12.6	7	9.5		4.7	18.3		
Complications in this pregnancy																				
	No	150	77.3			70.9	82.6	82	97.6			91.7	99.3	68	91.9		83.4	96.2		
	Yes	2	1			0.3	3.7	2	2.4			0.7	8.3	0						
	Missing	42	21.4					36						6						
Type of complication																				
	Placenta Previa	1	0.5			0.1	2.8	1	50			9.5	90.5							
	Edema	1	0.5			0.1	2.8	1	50			9.5	90.5							
Complication at birth																				
	No	143	73.7			67.1	79.4	77	93.9			86.5	97.4	66						
	Yes	5	2.6			1.1	5.9	5	6.1			2.6	13.5	0						
	Missing	46	23.7					38						8						
Type of complication																				
	Bleeding	1	0.5			0.1	2.8	1	20			3.6	62.4							
	Breech	1	0.5			0.1	2.8	1	20			3.6	62.4							
	Disproportion	1	0.5			0.1	2.8	1	20			3.6	62.4							
	Placenta retention	1	0.5			0.1	2.8	1	20			3.6	62.4							
	PROM	1	0.5			0.1	2.8	1	20			3.6	62.4							

**Table 2 ijerph-16-04170-t002:** Demographic information on the newborns included in the study.

Manin Variables		Area																		*p*-Value
		All locations						Inland Area (1)						Floating Area (2)						
		Count	Percent	Mean	SD	95% CI		Count	Percent	Mean	SD	95% CI		Count	Percent	Mean	SD	95% CI		
						Lower	Upper					Lower	Upper					Lower	Upper	
Babies																				
	Gestational Age	142		39.1	1	39	39.3	77		40.0	0.8	39.5	40.2	65		38.4	0.7	38.3	38.6	<0.0001
	Weight(g)	149		3089.6	401.6	3024.6	3154.6	83		3181.9	399.4	3094.7	3269.1	66		2973.5	375.9	2881.1	3065.9	0.001
	Length (cm)	147		48.9	2.9	48.5	49.4	83		48.5	3.4	47.8	49.3	64		49.5	2	49	49.9	0.06
	Head Circumference (cm)	146		31.5	2.3	31.1	31.9	81		32.6	1.2	32.3	32.8	65		30.2	2.6	29.5	30.8	<0.0001
	Ponderal Index (PI)	147		26.9	7.3	25.7	28.1	83		28.6	8.9	26.7	30.6	64		24.7	3.4	23.9	25.6	0.001
Apgar Score																				
	1 minute	130		7.3	0.8	7.2	7.4	82		7.3	0.9	7.1	7.5	48		7.4	0.6	7.5	7.2	0.54
	5 minutes	130		9.4	1.1	9.2	9.6	82		9.8	1	9.6	10	48		8.6	0.6	8.5	8.8	<0.0001
Mode of delivery																				
	Normal	134	69.1			62.3	75.2	68	81.9			72.3	88.7	66	89.2			80.1	94.4	
	Vacuum	8	4.1			2.1	7.9	8	9.6			5.0	17.9	0						
	Surgery	7	3.6			1.8	7.3	7	8.4			4.1	16.4	0						
	Missing	45	23.2					37						8	10.8					
Gender																				
	Male	81	41.7			35	48.8	58	56.9			47.2	66.1	23	32.9			23.0	44.5	
	Female	91	46.9			40	53.9	44	43.1			33.9	52.8	47	67.1			55.5	77.0	
	Missing	22	11.3					18						4						

**Table 3 ijerph-16-04170-t003:** Toxic exposing factors from the study areas. Results presented in percentages with 95% Confidence Intervals.

Exposing Factors	All Areas	Area 1	Area 2
Count	Percent	95% CI	Count	Percent	95% CI	Count	Percent	95% CI
Lower CI	Upper CI	Lower CI	Upper CI	Lower CI	Upper CI
Water												
	Tap water	1	0.5	0.1	2.9	1	0.8	0.1	4.6	69	93.2	85.1	97.1
	River & Pond	76	39.1	32.5	46.2	7	5.8	2.9	11.6	0			
	Rain Water	64	33.1	26.9	40.1	64	53.3	44.4	62.0	0			
	Well	45	23.2	17.4	29.2	45	37.5	29.4	46.4	0			
	Bottled	8	4.1	2.1	8	3	2.5	0.9	7.1	5	6.8	2.9	14.9
Insecticide Use												
	No	85	43.8	37	50.6	70	58.3	49.4	66.8	15	20.3	12.7	30.8
	Yes	109	56.2	49.2	63	50	41.7	33.2	50.6	59	79.7	69.2	87.3
Use Where												
	Home and Farm	36	33	24.9	42.3	35	70	56.2	80.9	1	1.7	0.3	9.0
	Home only	73	67	57.7	75.1	15	30	19.1	43.8	58	98.3	91.0	99.7
	Missing Data	85				70				15			
Fishing												
	No	65	33.5	27.2	40.4	62	51.7	42.8	60.4	3	4.1	1.4	11.3
	Yes	129	66.5	59.6	72.8	58	48.3	39.6	57.2	71	95.9	88.7	98.6
Fishing location												
	Lake	47	37.0	29.1	45.7	47	83.9	72.2	91.3	71	100	94.9	100
	Stream/River	9	7.1	3.8	12.9	9	16.1	8.7	27.8	0			
	Large River	71	55.9	47.2	64.2	0				0			
	Missing Data	67				64				3			
Food Source												
	In Village	123	63.7	56.7	70.2	119	99.2	95.4	99.85	4	5.5	2.2	13.3
	Market	70	36.3	29.8	43.3	1	0.8	0.1	4.57	69	94.5	86.7	97.8
	Missing Data	1								1

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
