# Peer review of "An Increased Risk of Stunting among Newborns in Poorer Rural Settings: A Cross-Sectional Pilot Study among Pregnant Women at Selected Sites in Rural Cambodia"

_ijerph, 2019, doi:10.3390/ijerph16214170_

Round 1

Reviewer 1 Report

I appreciate a lot the manuscript and is not easy to give some suggestions because the paper is really well done

Only two considerations:

1. What do similar studies but non-rural areas highlight? can you discuss it better in the discussion?

2. what approach of public health advice as a result of your experience?

Author Response

Please see the attachment with response to all reviewers

Reviewer 2 Report

This is a paper introducing an interesting study about exposure to Persistent Toxic Substances (PTS) and mothers and newborn anthropometric features. In the present paper only the socioeconomic data is described and not data about PTS levels is presented. The main interest of this study are the PTS levels and there possible association between those and stunting, without these results this paper lack of much scientific interest.

In this paper the authors present the design and the descriptive characteristics of the studied population and there are relevant results. Differences between the two compared groups of mums and children from two regions, not too far apart though, have not much scientific interest. The different in the results from the analytical results of the substances tested in blood are far more interesting but the authors didn’t present those results in the present paper, so this paper is quite doesn’t have much to add

Reviewer 3 Report

Dear authors

my compliments for your field work in conditions that are unlikely to be easy.

Your manuscript refers to a descriptive study that appears preparatory to a second, more analytical work and this certainly arouses interest in future study but limits it to the present.

The reduced sample size is plausibly due to operational difficulties, which should be made explicit.

Please check the following questions and suggestions:

1. Introduction

P.2 lines 61-64: The descriptive objective is too generic, the rationale is not clearly reported (does the study investigate differences between areas or the study is poorly descriptive?)

2. Materials and Methods

P.2 line 67: please explain what means “compatible with”

P.2 line 70: the sentence “this paper describes the framework for the study” means that is a preparatory description in the view of a real study ?

2.3 Inclusion

P.3 lines 98-106: why were the numbers so small? Why the authors do not try to check the powerfull of the sample size in the groups comparisons ?

2.5 Sampling and analyses

P.3 line 118: how the samples planning was decided ?

2.6 Ponderal index

P.4 lines 131-134: references are needed to support the statement

2.7 Statistical model

P.4 lines 145-146: t-test is the right test for comparing between two means, why also ANOVA ?

the table 3 is recalled twice (line 214 and 308) and commented but it is not reported in the manuscript nor in the supplementary material

Reviewer 4 Report

The article discusses the important issue of the influence that environmental factors exert on the anthropometric parameters of women (future mothers), as well as on the course of pregnancy of these women and the size of newborns. This is an interesting research topic, especially in countries struggling with poverty.

This study covers only a few factors that may impact the indices measured (it does not include, for example, chewing tobacco during pregnancy which is popular in Cambodia), but after an extensive correction, the article may be accepted for publication.  

Specific comments:

Due to a limited population under examination, as well methodological defects and the fact that the study is the first of its kind in Cambodia, as the authors have noted, I propose to add the following wording to the title of the article: "A pilot study"

Introduction should be limited to matters directly related to the subject of the study. In line 56, the abbreviation (AMAP) should be added following the words "...Arctic Monitoring and Assessment Programme", as it is used later in the article, but it is not explained anywhere else.

The whole text of the article does not contain a uniform number of examined patients. In line 98 (part of Materials and Methods), 194 women are indicated, including 120 women from Chroy Sdao (area 1) and 74 from Eak Phnom (area 2), while Tables 1 and 2 mention 193 women, including 119 from area 1 ??  

In point 2.5 (line 119) it is stated that blood and urine samples were taken to be tested in the population of women under study, but in the whole article there is no information what was marked in them and what were the results!!!! If these factors are not taken into account in the article submitted for review, the information should be removed. The same applies to the abstract of the article.

Tables 1 and 2 are incomprehensible and require an extensive correction. First of all, a column with the p-value should be added, comparing the two examined groups; this because the authors repeatedly state that the examined parameters differed significantly between the groups, but the p-value was not indicated anywhere. In addition, the number of newborns per group should be given in Table 1. From line 106 it follows that 146 newborns were examined, including 82 from groups 1 and 64 from group 2. Table 1 in its current shape suggests that there were as many newborns as mothers, i.e. 193 ??

       Tables 1 and 2 have the same title, and in addition to the title of table 2,

        the authors write (continued)?

       Likewise, in Table 2 the numbers of women examined do not tally. For example, if you add up women with different educational backgrounds in group 1, you get the number 107, and similar differences are found in other parts of this table. The same applies to the percentage of women which does not add up to 100%.

Figures 2, 3, 4, 5 and 6 are completely unnecessary, as the data presented in them are included in Tables 1 and 2, and additionally given in the text of the article. These figures do not represent anything new to the reader.

In lines 179 and 201, 40 weeks of pregnancy are mentioned, whereas in Table 1, it is 39.7 weeks ? Line 184 mentions 149 deliveries, and line 106 gives 146 neonates?  Line 187 reads as follows: 'No complications were reported from the 66 deliveries in area 2', whereas Table 2 states:  'No complications in this pregnancy' for 68 women. According to the information given in line 106, there were 64 women in group 2 ?

In point 3.1.2. Neonates (lines 200-201), the information which has been already provided in verses 178-179 is repeated.

The submitted text of the article does not include Table 3, referred to in point 3.2. Possible sources of exposure to persistent toxic substances??

In line 265, when indicating the new-born’s body weight, length and head circumference, do the figures in brackets 500 g, 1,8 cm and 1,3 cm indicate the standard deviation SD - if this is the case, ± SD should be added.

Round 2

Reviewer 2 Report

Since the authors are not going to include any results of association between the PTS levels and stunting. “Possible associations between stunting and exposure to 24 Persistent Toxic Substances (PTS) will be investigated in future studies.” To avoid misunderstandings references to PTS must be removed from the abstract and the title must be more illustrative of the conclusion of the study that is that socio-economic factors could be associated with the significant differences between women and their newborns from two separate study sites. The discussion can include that this association can also be due to differences in the PTS levels and this is going to be investigated in future studies. With the current tittle and abstract the potential readers expect to find results regarding PTS levels not associations with socio-economic factors, that are the only ones analyses so far.

Author Response

Please find the attached reply to all reviewers.

Reviewer 3 Report

Dear authors

I would like to congratulate you on your extensive review work.
Since you have fully or sufficiently replied to the remarks addressed to you, as far as I am concerned, your paper can be published as a pilot study of general interest.

all the best

Reviewer 4 Report

Still there is information about analysis of blood and urine (in the article and in the abstract) although in response to the review authors are writing that this information was removed ?
